# Engineered nanointerfaces for microfluidic isolation and molecular profiling of tumor-specific extracellular vesicles

Eduardo Reátegui[1,2,3,4,5,11], Kristan E. van der Vos [6,12], Charles P. Lai[6,13], Mahnaz Zeinali[1,2,3,4], Nadia A. Atai[6], Berent Aldikacti[1,2,4], Frederick P. Floyd Jr.[1,2,3], Aimal H. Khankhel[1,2,3], Vishal Thapar[4], Fred H. Hochberg[7], Lecia V. Sequist[4,8], Brian V. Nahed[4,9], Bob S. Carter[7], Mehmet Toner[1,2,3,5], Leonora Balaj[6], David T. Ting[4], Xandra O. Breakefield[6,10] & Shannon L. Stott[1,2,3,4,5,8]

Extracellular vesicles (EVs) carry RNA, DNA, proteins, and lipids. Specifically, tumor-derived EVs have the potential to be utilized as disease-specific biomarkers. However, a lack of methods to isolate tumor-specific EVs has limited their use in clinical settings. Here we report a sensitive analytical microfluidic platform ([EV]HB-Chip) that enables tumor-specific EV-RNA isolation within 3 h. Using the [EV]HB-Chip, we achieve 94% tumor-EV specificity, a limit of detection of 100 EVs per μL, and a 10-fold increase in tumor RNA enrichment in comparison to other methods. Our approach allows for the subsequent release of captured tumor EVs, enabling downstream characterization and functional studies. Processing serum and plasma samples from glioblastoma multiforme (GBM) patients, we can detect the mutant EGFRvIII mRNA. Moreover, using next-generation RNA sequencing, we identify genes specific to GBM as well as transcripts that are hallmarks for the four genetic subtypes of the disease.

[1] Center for Engineering in Medicine, Massachusetts General Hospital, Harvard Medical School, Charlestown, MA 02129, USA. [2] Harvard Medical School, Boston, MA 02114, USA. [3] Department of Surgery, Massachusetts General Hospital, Harvard Medical School, Boston, MA 02114, USA. [4] Massachusetts General Hospital Cancer Center, Harvard Medical School, Boston, MA 02114, USA. [5] Shriners Hospital for Children, Harvard Medical School, Boston, MA 02114, USA. [6] Neurodiscovery Center, Massachusetts General Hospital, Harvard Medical School, Boston, MA 02124, USA. [7] Department of Neurosurgery, University of California San Diego, La Jolla, CA 92093, USA. [8] Department of Medicine, Massachusetts General Hospital, Harvard Medical School, Boston, MA 02114, USA. [9] Department of Neurosurgery, Massachusetts General Hospital, Harvard Medical School, Boston, MA 02124, USA. [10] Department of Neurology and Radiology, Massachusetts General Hospital, Harvard Medical School, Boston, MA 02114, USA. [11]Present address: William G. Lowrie Department of Chemical and Biomolecular Engineering, Comprehensive Cancer Center, The Ohio State University, Columbus, OH 43210, USA. [12]Present address: Department of Molecular Carcinogenesis, Netherlands Cancer Institute, 1066 CX, Amsterdam, The Netherlands. [13]Present address: Institute of Atomic and Molecular Sciences, Academia Sinica, Taipei 10617, Taiwan. Eduardo Reátegui, Kristan E. van der Vos, and Charles P. Lai contributed equally to this work. Correspondence and requests for materials should be addressed to S.L.S. (email: sstott@mgh.harvard.edu)

Extracellular vesicles (EVs) carry proteins, mRNAs, micro-RNAs, other non-coding RNAs, DNAs, and lipids, serving as an endogenous delivery vehicle for cell-to-cell communication[1]. Tumorigenesis affects many pathways regulating the production of EVs resulting in an increased production of EVs by some tumor cells in comparison to normal cells[2]. These tumor EVs contain a select subset of proteins and nucleic acids that can manipulate their cellular microenvironments at local and distant sites to promote angiogenesis, invasiveness, and metastasis[3–5]. Cancer patients frequently show increased concentrations of EVs in their circulation[6,7] that allows the use of isolated EVs from biofluids as biomarkers for diagnostics and disease monitoring in a much-needed non-invasive manner.

EVs have not been widely applied in clinical settings due to current limitations in EV isolation technology that mainly rely on EV physical properties using ultracentrifugation and precipitation processing. These two techniques isolate not only tumor EVs, but also EVs derived from non-malignant cells such as platelets, endothelial cells, and immunological cells, yielding low-throughput outcomes and specificity. Different protocols have been described to isolate tumor EVs using antibody-coated beads, and silica substrates[8,9]. However, bead-based assays take a relatively long time and consist of multiple labeling steps[9–11]. Our group and others have used various microfluidic approaches for fast and reproducible immunoaffinity isolation of tumor EVs from biofluids[12]. Nonetheless, the majority of these approaches target tetraspanins and annexins, ubiquitous proteins present on the surface of the majority of EVs to capture them;[12–14] or use anti-EpCAM antibodies that are also expressed on epithelial cells[15], thereby limiting the specificity of the isolated tumor EVs. Other microfluidic strategies, such as deterministic lateral displacement (DLD), have been developed to sort populations of small nanometer EVs from micrometer-size particles[16]. Recently, a nano-DLD device has achieved separations between 10 and 110 nm populations of exosomes[16]; despite its sorting resolution on the size of the vesicles, the method lack of specificity toward tumor-specific EVs and may miss detection of important biological cargo. Other approaches include the use of plasmonic sensor devices that can immobilize and then quantify EVs with improved sensitivity. However, these devices are complicated to manufacture and scale up, and usually, operate at low throughput[17–19].

For this study, we integrate our herringbone microfluidic device, an immune-affinity based, a high-throughput technology initially used for rare cell isolation, with a thermally responsive nanostructured substrate that provides further enhancement of tumor-specific capture sensitivity (EVHB-Chip)[20]. The nanostructured substrate consists of an ultra-thin (~150 nm) gelatin membrane functionalized with streptavidin-coated nanoparticles that when combined with the chaotic mixing resulting from the herringbone grooves, maximizes EV interactions with the tumor-specific antibody-coated surfaces. We engineer optimal configurations for the surface-immobilized antibodies by using different nanometer-sized PEG linkers that decreased the $Z_{potential}$ of the EVs, capture antibody, and linker complex formed such that tumor-specific EV capture is enhanced. The immobilized tumor-specific EVs were quantified by confocal microscopy and quantitative PCR. Furthermore, the microfluidic platform was manufactured with cyclic olefin copolymer (COC) that ensures scalability and sample reproducibility while minimizing production cost and enabling clinical trials in multiple cancer centers[21].

In this report, we show that our EVHB-Chip can efficiently capture tumor-derived EVs from plasma and serum. Functionalization of the EVHB-Chip with a cocktail of antibodies allows for specific and rapid isolation of tumor-derived EVs that can subsequently be released allowing for the analysis of their biological cargo. The design with an optimized PEG linker that immobilizes antibodies to the nano-structured interface results in over 10-fold increase in tumor-EV enrichment ratios when compared to gold-standard methods[22,23]. Next, we demonstrate the clinical potential of our device by identifying the EGFRvIII mutation in serum/plasma from GBM patients. Importantly, RNA sequencing analysis on the EVs reveals the presence of more than 50 genes specific to GBMs, as well as a variety of the GBM subtype-identifying mRNAs (neural, pro-neural, mesenchymal, and classical).

## Results

**Microfluidic workflow for the isolation of tumor EVs.** To enrich tumor-specific EVs from the serum or plasma of cancer patients, we developed a microfluidic platform that was based on antigen-specific capture of the tumor vesicles. While our approach requires the identification of tumor-specific antigens present on the EV surface, the "positive selection" of tumor EVs results in a sample enrichment that enables deeper and more complex downstream analysis of the molecular and genetic content of the tumor EVs. To promote interactions between the EVs and the antibody-coated surface of the microfluidic device, a staggered herringbone mixing pattern was used[24] to create anisotropic flow as the fluid travels through the device. This flow pattern results in micro-vortices that promote the number of interactions between the EVs and the antibody-coated surface thereby enhancing binding of tumor EVs to the chip. Moreover, the small inner volume of the device (~90 μL) enables the processing of a wide range of sample volumes (100 μL to 5 mL).

The resulting microfluidic device, EVHB-Chip, was manufactured from cyclic olefin copolymer (COC) using a micro-injecting molding process (Fig. 1a), enabling mass production of the device, while reducing material autofluorescence and maximizing biocompatibility. The design evolved from our original poly-dimethylsiloxane (PDMS) herringbone chip[25] and comprised two different layers that were thermally bonded: the first layer, consists of a 3D herringbone structure pattern; the second layer of the device was a flat Topaz coupon of a thickness similar to a coverslip to facilitate confocal microscopy analysis. The feature sizes of the herringbone structures were selected as: overall height of the channel 50 μm; the ratio of the height of the grooves to the height of the channel was set to 0.8; the angle between the herringbones and the axis of the channel was 45°; and the principal wave vector was $q = 2\pi/100\ \mu m$[24–26]. To operate the device, a simple lab-grade syringe pump is required (see Methods), with the syringe connecting to the device using a standard luer lock connection. The total footprint of the device replicates a 1″x3″ glass slide.

**Antibody selection and engineering of fluorescent EVs.** Using data in the literature that identified surface markers highly expressed in glioblastoma cell lines, EVs and primary tumors[27], antibodies directed against EGFR, EGFRvIII, ephA2, podoplanin, PDGFR, and MCAM were screened for specificity and sensitivity with our microfluidic device. Then, we functionalized the EVHB-Chip with individual or multiple tested antibodies using a silane surface chemistry (zero-length spacer) to capture tumor EVs. For these experiments, cell lines and fluorescent EVs spiked into human control serum were used to identify the most promising tumor EV-specific antibody candidates (Supplementary Fig. 1). Human Gli36 glioma cells were engineered to produce fluorescently labeled EVs to enable rapid, visual quantification of tumor-specific EVs during microfluidic optimization steps. We infected Gli36 wild type (Gli36wt), which express EGFR, and Gli36 cells stably expressing EGFRvIII (Gli36-EGFRvIII) as well as EGFR

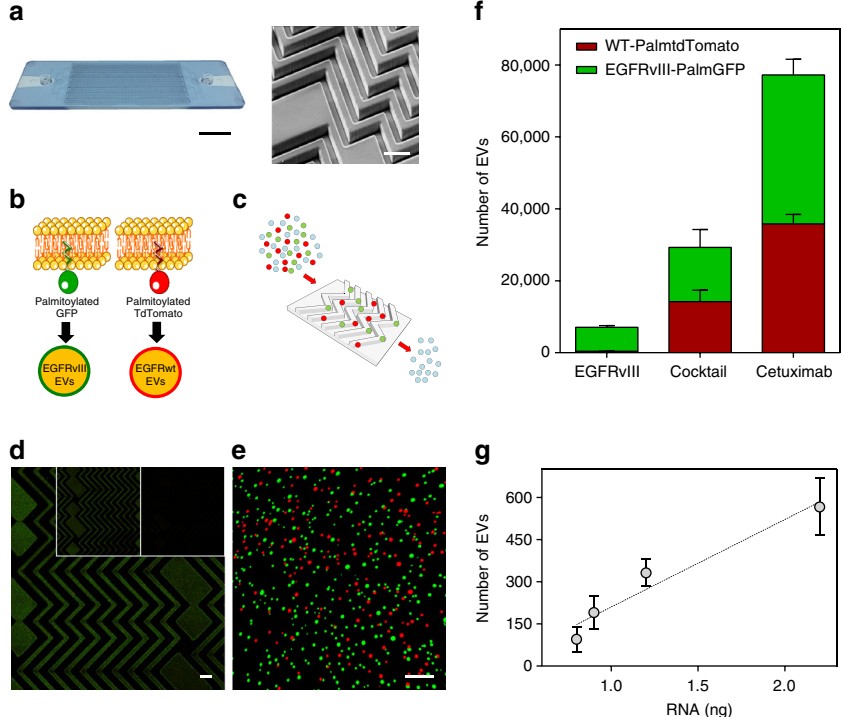

**Fig. 1** EVHB-Chip design and reporter constructs. **a** Image of the operating COC EVHB-Chip. Pressure-driven flow-pumped serum or plasma through an inlet at 1 mL h$^{-1}$. Waste serum or plasma is collected in a single outlet (scale bar 1 cm). On the right, a SEM micrograph of the 3D herringbone features of the microfluidic device (scale bar 100 µm). **b** Gli36wt or Gli36-EGFRvIII glioma cells were stably infected with lentivectors encoding palmitoylated-tdTomato (PalmtdTomato) or -GFP (PalmGFP), respectively, to produce fluorescently labeled EVs. **c** Serum or plasma from healthy donors with spiked-in fluorescent EVs (in red and green) was run through the microfluidic device coated with tumor-specific antibodies to capture tumor EVs. **d** A collage of one hundred confocal microscopy images of fluorescent EVs captured on the surface of the device. Different antibodies were used: large image (Cetuximab, scale bar 50 µm); right insert (antibody cocktail); left insert (anti-EGFRvIII). **e** Representative image of digitally rendered signals of captured EVs shown as green (EGFRvIIIPalmGFP EV) or red spheres (WT-PalmtdTomato EV; scale bar 1 µm). **f** Quantification of captured wild-type PalmtdTomato and EGFRvIII PalmGFP EVs on the microfluidic device when different antibodies were used. A concentration of 20 µg mL$^{-1}$ were used for EGFRvIII and Cetuximab, and 10 µg mL$^{-1}$ for each antibody present in the cocktail ($n = 3$ technical replicates; ±standard error of the mean, s.e.m.). **g** After imaging captured EVs were lysed. Total RNA was isolated and correlated to confocal microscopy data ($n = 3$ technical replicates; ±s.e.m.)

with lentiviral vectors encoding PalmtdTomato or PalmGFP, respectively (Fig. 1b)[28]. The fluorescent tumor EVs derived from these cells were then spiked into human plasma or serum and processed through the functionalized EVHB-Chip (Fig. 1c). Captured EVs were visualized by confocal microscopy and demonstrated the tunability of the platform depending on the antibody of choice. We tested the specificity of the capture by analyzing the binding of the mixed population of EVs and observed that the EGFRvIII antibody mostly captured the green-fluorescent EVs expressing EGFRvIII, while chips coated with an antibody recognizing both EGFRwt and EGFRvIII (Cetuximab) captured both green- and red-fluorescent EVs. For EGFRvIII-specific antibody capture, the specificity of the device was 94.3 ± 0.6% ($n = 3$ technical replicates, mean ± s.e.m.) (i.e., EGFRvIII-PalmGFP EVs). Moreover, an antibody cocktail directed against EGFR, EGFRvIII, podoplanin, and PDGFR also exhibited capture affinity for both populations of EVs (Fig. 1d–f, Supplementary Fig. 2). When using EVs that originated from parental cell lines with overexpression of EGFR and EGFRvIII (i.e., Gli36), our data showed a higher enrichment of tumor-specific EVs when Cetuximab alone was used (20 µg mL$^{-1}$ used on-chip, Fig. 1f). For the cocktail of antibodies used in Fig. 1f, the amount of Cetuximab bound on-chip was reduced by a factor of two (10 µg mL$^{-1}$) to allow for efficient placement of each antibody in the mixture. The resulting tumor-specific EV enrichment for Gli36 EVs was reduced by the same amount. When EVs from a cell line without EGFR overexpression was used (i.e., GBM 20/3), the cocktail of

antibodies outperformed Cetuximab only (Supplementary Fig. 2). Considering that the majority of assays for isolated EVs are molecular-based, we compared the number of EVs imaged on the microfluidic device with the total mass of RNA extracted from the chip. The linear correlation between the two measurements ($r^2 = 0.83$, Fig. 1g) validated that our imaging-based approach was a reasonable surrogate for RNA-based analysis for efficient optimization during the development of the final chip design.

**Optimization of nanointerface design**. It is well known that antibody-based capture in microfluidic devices can be sensitive to processing conditions and as such, we explored the impact of these conditions on EV isolation using the EVHB-Chip. First, the impact of flow rate was evaluated using tumor cell-derived EVs spiked into human serum or plasma and run on the EVHB-Chip at 0.5, 1, 2, and 5 mL h$^{-1}$. As demonstrated in Fig. 2, varying the flow rate of the fluid changed the number of captured EVs. At flow rates below 1 mL h$^{-1}$, more EVs were captured, as indicated by a high amount of RNA obtained (24.3 ± 2.3%). As the flow rate was increased to 2 mL h$^{-1}$ or higher, the total amount of RNA dropped by 88% (Fig. 2a). To quantify if these total RNA amounts were specific to the isolation of tumor EVs or non-specific binding of EVs to the surface, we analyzed the tumor-specific EV transcript (EGFRvIII) and a general EV transcript (glyceraldehyde 3-phosphate dehydrogenase, GAPDH) using a Taq-Man® Gene Expression assay, resulting in our enrichment ratio

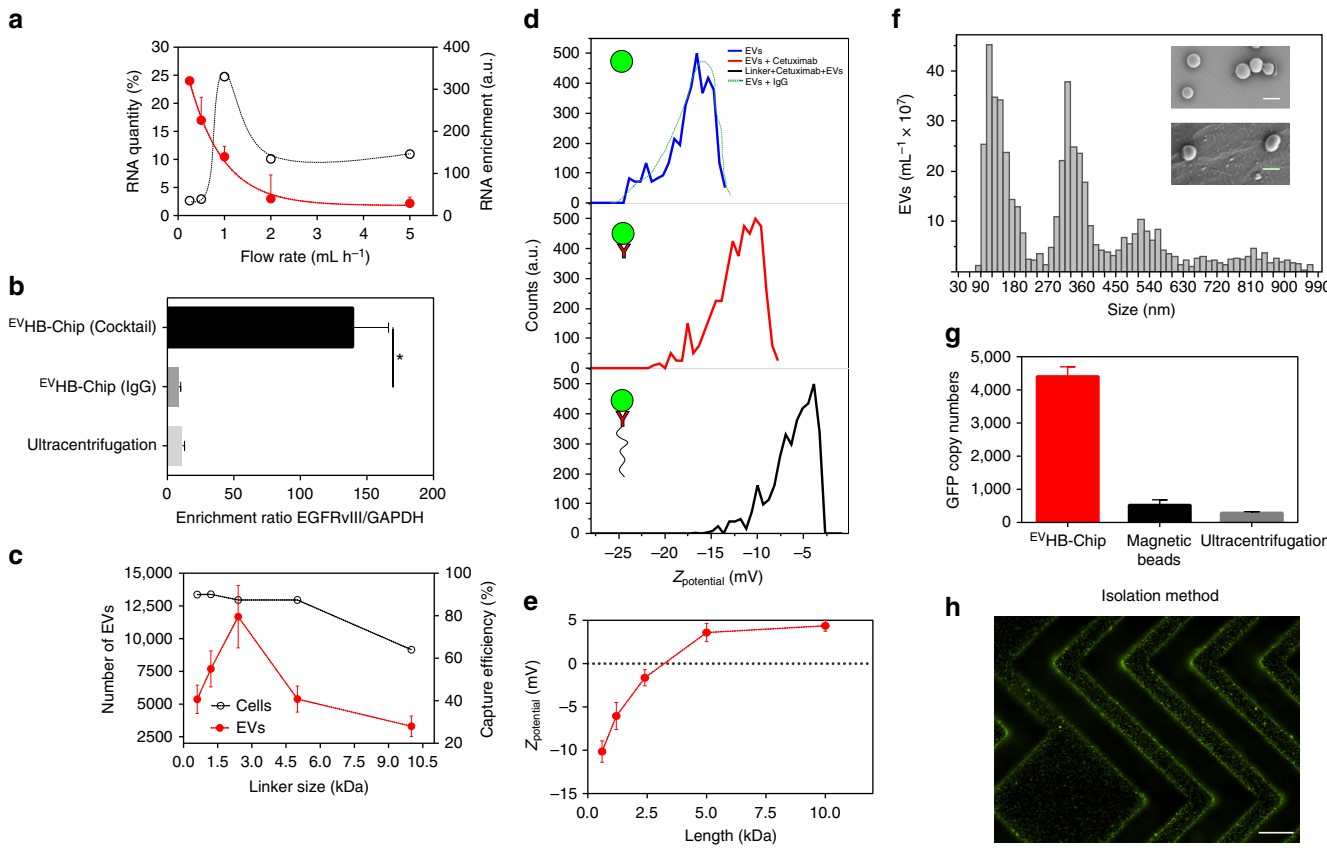

**Fig. 2** Characterization and operation of the COC $^{EV}$HB-Chip. **a** RT-PCR quantification of the RNA recovered from EVs captured on the microfluidic device coated with Cetuximab at different flow rates. RNA enrichment ratio in the microfluidic device was calculated between EGFRvIII and GAPDH and depicted by the dotted black line. The red line represents the total RNA quantity. **b** Comparison of RNA enrichment ratios between ultracentrifugation and different capture antibodies on the surface of the $^{EV}$HB-Chip (*$p < 0.05$, one-way ANOVA). **c** Effect of linker size to captured tumor-specific EVs and cells at 1 mL h$^{-1}$. **d** Changes in $Z_{potential}$ distribution for different complex configurations. EVs, EVs binded to antibodies, and EVs binded to linked antibodies. **e** Variation in $Z_{potential}$ for different PEG linker size of the complex EV and antibodies. **f** Size and concentration of tumor EVs isolated with the microfluidic device. Different size distributions were identified within the range of 40–1000 nm. Insert shows oncosomes (top, scale bar 2 μm) and exosomes (bottom, scale bar 100 nm). **g** Performance comparison between the $^{EV}$HB-Chip and standard EV isolation methods. Plasma from healthy donors with spiked-in fluorescent EVs isolated from Gli36-EGFRvIII cells was run through the microfluidic device coated with Cetuximab to capture tumor EVs. In parallel, tumor EVs were isolated using magnetic beads coated with Cetuximab or by ultracentrifugation. ddPCR was used to quantify the total number of EGFRvIII copies captured with the different techniques. **h** Fluorescent microscopy image of immunoaffinity stained tumor-specific EVs on the surface of a microfluidic chip (scale bar 50 μm). For **a**, **b**, **c**, **e**, and **g** $n = 3$ technical replicates; ±s.e.m. For **d** and **f**, a representative experiment is shown from $n = 3$ technical replicates

metric. While flow rates <1 mL h$^{-1}$ resulted in the maximum yield of RNA, the highest enrichment ratio for tumor-specific RNA was 340 at 1 mL h$^{-1}$ (Fig. 2a). A flow rate of 1 mL h$^{-1}$ allows for 1 mL of plasma to be processed through the entirety of our assay (inclusive of all wash steps and RNA extraction) in less than 3 h. The performance of the $^{EV}$HB-Chip was then compared against bulk EV analysis (i.e., ultracentrifugation) of the input samples and eluates, or "waste" of the chip. From this analysis, the $^{EV}$HB-Chip demonstrated a remarkable tumor-EV capturing specificity with more than a 10-fold enrichment of EGFRvIII transcripts (Fig. 2b, Supplementary Fig. 1). Additionally, the specificity of the enrichment was tested at different dilutions, still showing a better performance than ultracentrifugation (Supplementary Fig. 3).

Our initial experiments for the targeted capture of tumor EVs utilized standard microfluidic surface chemistry that has been highly successful for the isolation of biological cells from complex fluids[25]. Early data evaluating capture efficiency for glioblastoma cell lines and tumor-specific message for EVs isolated from the same cell lines suggested that under the same processing conditions, the quantity of tumor-specific message of EVs had a dependence on antibody concentration and processing dilution

(Supplementary Figs. 1 and 2). Therefore, recognizing that the biophysics of capture were different for the ~100 nm diameter EVs in comparison to whole cells (10–20 μm), we tested different antibody immobilization strategies at the surface of the $^{EV}$HB-Chip to improve capture of EVs. First, a zero-length spacer (I) immobilized directly on the surface of the device; second, a zero-length spacer immobilized on a nanostructured substrate (II)[20]; and third, different nm-length PEG spacers were used (III; Supplementary Fig. 4). Configurations I and II were previously used for immunoaffinity capture of μm-size particles (e.g., circulating single or clusters of tumor cells)[20]. When PEG spacers were combined with the nanostructured substrate (Supplementary Fig. 5), we achieved an approximately two-fold increase in the capture of EVs (Supplementary Fig. 4, $p < 0.05$, one-way ANOVA).

We further explored the effect of the length of separation of the antibodies to the nanostructured substrate. We noticed that for micrometer-sized cells, the size of PEG spacers is almost negligible regarding capture efficiency (Fig. 2c). For sizes ranging from 0.6 to 5 kDa the capture efficiency cells ranged from 93 ± 2.4 to 92 ± 1.8%, respectively; with a decrease in capture efficiency to 63 ± 2.1% when the length was increased to 10 kDa. Surprisingly,

a different profile of capture efficiency was observed for EVs. For PEG linkers between 0.6 and 2.4 kDa, there was a proportional increase in the number of Gli36wt PalmtdTomato EVs captured on the surface of the microfluidic device. However, PEG spacer with a molecular weight ($M_w$) >2.4 kDa showed a decrease in the number of captured EVs (e.g., 5, 10, 15, 20 kDa). Further experiments shown that EVs binded to an antibody linked to a PEG chain decrease significantly the $Z_{potential}$ of the total complex formed likewise was an indication of specific binding ($n = 5$, Fig. 2d, Supplementary Fig. 5); with a PEG linker of 2.4 kDa showing a proximity to zero $Z_{potential}$ ($n = 4$, Fig. 2e).

**Development of optimal processing conditions.** Next, we evaluated the ability of the $^{EV}$HB-Chip to capture tumor-specific EVs without size-bias. First, a mixture of biotinylated-nanoparticles of different sizes was spiked in plasma ($5 \times 10^9$ particles per mL) and was run through the device at a flow rate of 1 mL h$^{-1}$. We found that regardless of size, particles were captured across the length of the device with no significant differences ($p > 0.05$, one-way ANOVA, Supplementary Fig. 6). We also characterized the size distribution of EVs isolated from Gli36 cells pre- and post-$^{EV}$HB-Chip capture by a tunable pulse resistive sensing (TPRS) method and found that different populations of EVs were present (Fig. 2f).

We further explored ways to increase the capture of tumor-specific EVs. We noticed that running EVs or cells first through a chip coated with control IgG before running through the $^{EV}$HB-Chip increased specific capture by 15%. This increase in capture was not dependent on the antibody in the first chip since a blank chip also increased specific capture in the second chip. We performed additional experiments to show that the IgG chip does not deplete the tumor-specific EVs by quantifying the changes in EV concentration before and after flowing EVs spiked in PBS through the device. We found that <5% of the EVs remained on the IgG-coated device. Additionally, PCR analysis indicated that the IgG-coated device captured 10-fold fewer EVs in comparison to the $^{EV}$HB-Chip coated with tumor-specific antibodies (Supplementary Fig. 7).

Next, we analyzed whether the surface of the $^{EV}$HB-Chip was saturated by running an EV sample through two $^{EV}$HB-Chips in series. We noticed depletion on the number of EVs between the first and the second device at different dilutions of the sample, suggesting that the first $^{EV}$HB-Chip was saturated (Supplementary Fig. 8). Next, we tested an experimental setup with five $^{EV}$HB-Chips in series to increase the total amount of tumor-specific EVs captured on the $^{EV}$HB-Chip surface. When we analyzed each chip individually, we observed the highest capture efficiency in the first $^{EV}$HB-Chip and a gradual decrease in capture in the subsequent devices. The GFP/PPBP enrichment ratio was relatively stable over the first four $^{EV}$HB-Chip in series, indicating that tumor EVs were captured with relatively high specificity. However, we observed a significant drop of 92% of the GFP/PPBP enrichment ratio in the 5th $^{EV}$HB-Chip ($p < 0.05$, Student's $t$-test, between 4th and 5th device, Supplementary Fig. 8). These optimization studies using serum or plasma samples with spiked in tumor EVS showed us that most efficient capture of tumor EVs was obtained when we used four $^{EV}$HB-Chips in series preceded by a blank chip. Therefore, we used this experimental setup for the capture of tumor EVs from clinical samples.

Due to increasing interest in functional studies with EVs[4], our platform allows different techniques for recovering captured EVs. First, EVs can be eluted from the surface of the $^{EV}$HB-Chip by flushing a proteinase K solution (0.05 %) that shaves EVs from the device. Second, a thermally responsive gelatin nanocoating[20]

was selected as the base layer for nanoparticles to be attached to the $^{EV}$HB-Chip. At room temperature (RT), the gelatin nanocoating is highly stable; when the temperature of the device is raised to physiological temperature (37 °C), the coating dissolves within seconds, releasing all captured EVs[20]. To test the success of EV release using proteinase K captured EVs were released from the surface of the device (see Methods) and subsequently quantified by confocal microscopy (Supplementary Fig. 9).

Following optimization of the microfluidic platform at three different levels (e.g., device processing conditions, capture antibodies, releasable nanointerface coating), we calculated the capture efficiency and the limit of detection of the $^{EV}$HB-Chip. For EV capture efficiency, we spiked in a known number of EVs (concentrations between 35 and 50 million of particles per mL of PBS were used) and flowed the solution through the chip. The concentration of EVs was analyzed before and after the samples were run through the $^{EV}$HB-Chip to determine how many EVs were depleted from the sample. Our results indicate a capture efficiency of 58.77 ± 5.37 (mean ± s.e.m., Supplementary Fig. 10) for the $^{EV}$HB-Chip. The limit of detection was calculated using the average fluorescence intensity of EVs[10,14]. For these experiments, we prepared a series of dilutions of spiked EVs in PBS from a known stock concentration. Then, the EVs captured on the $^{EV}$HB-Chip were imaged on a fluorescent microscope with the same exposure times for all the different titration conditions. It is important to mention that we do not measure individual EV fluorescence; the nanoparticle layer deposited on the surface of the device aggregates the EVs on the surface allowing them to be visualized and quantified at a bulk level. The aggregation of EVs has been previously shown to be appropriate to quantify limits of detection[10,14]. We found that our current limit of detection was 100 EVs per μL (Supplementary Fig. 11).

We tested the final design of the $^{EV}$HB-Chip against standard methods of EV isolation. Ultracentrifugation and magnetic bead separation are widely used for the isolation of EVs[29]. Samples of PalmGFP-EGFRvIII GBM EVs spiked into plasma were divided in triplicates and run independently on each platform. For ultracentrifugation samples were centrifuged for 2 h at $100,000 \times g$; for magnetic separation, samples were incubated with 3 μm magnetic, antibody-coated, polystyrene beads for 2 h[30]; and the $^{EV}$HB-Chip were run for the same amount of time (see Methods). For all three platforms, isolated EVs were lysed and homogenized with 700 μL of Qiazol buffer. Quantification of the tumor EV-specific message (GFP) indicated that the microfluidic platform had a 10-fold higher GFP mRNA content compared to ultracentrifugation and bead-based separation methods (Fig. 2g).

To characterize the benefit gained from using 3D herringbone structures in our chip design, we compared tumor-specific EV capture in a flat channel microfluidic device vs. a device of the same dimensions but with staggered herringbone groves in the ceiling of the device (all other parameters held constant, see Methods). Our data demonstrate that the addition of herringbone structures results in the capture of ~60% more EVs (Supplementary Fig. 12). We also calculated the shear stress that was most favorable for antibody binding of EVs at different flow rates. We used a Hele–Shaw device (Supplementary Fig. 12) in which the shear stress along the axis of the chamber decreases linearly along the chamber length without changing the input flow rate[31–33]. At 1 mL h$^{-1}$, the shear stress experienced by the EVs was 0.11–1.1 dyn cm$^{-2}$ with a drop of 19.6% in the fluorescence intensity of captured EVs, between the lowest to highest points of shear stress. When the flow rate was increased to 3 mL h$^{-1}$, the resulting shear stress was calculated to be between 0.32 and 3.17 dyn cm$^{-2}$ with a 92.8% decrease in the fluorescence intensity, indicating a marked

decrease in EV capture at these shear stresses (Supplementary Fig. 12).

Not all EV assays are based on molecular analysis, and it is important to highlight that the size and optical transparency of our $^{EV}$HB-Chip is highly suitable for visualization of EVs by immunoaffinity staining on-chip. To accomplish this, EVs from a human glioma cell line Gli36 were captured on the $^{EV}$HB-Chip and subsequently labeled using an anti-EGFR antibody and a fluorescent secondary antibody (see Methods). We were able to visualize the fluorescent signal produced by the captured EVs at the surface of the device (Fig. 2h).

**Molecular profiling of tumor EVs from glioblastoma patients.** In glioblastoma, the high degree of intra-tumor heterogeneity can complicate the genetic analysis of biopsy samples, and important oncogenes like EGFRvIII which promotes tumor formation by activating aberrant signaling and epigenetic pathways[34], can have a variable expression pattern within the tumor[35]. Therefore, tumor EVs released into the blood stream may provide a more accessible and representative source of biomarkers, potentially providing real-time information regarding tumor response and subsequent evolution in response to treatment. For this study, a group of 13 patients and 6 healthy donors was used to test the clinical utility of the $^{EV}$HB-Chip (processing details provided in Supplementary Fig. 13). In parallel experiments, tissue and cerebrospinal fluid (CSF) biopsies of six patients were performed for molecular profiling, including positive evaluation for the EGFRvIII mutation in three of six of the tumor samples. The biopsies and CSF samples showed a significant variability in the EGFRvIII analysis, with only one patient (P3) demonstrating EGFRvIII positivity in both CSF and tumor tissue (Supplementary Tables 1 and 2).

EVs were isolated from fresh and banked GBM patient serum ($n = 2$) or plasma ($n = 11$) using the $^{EV}$HB-Chip, with 2 mL of sample tested for each patient. To gain insight into the capture efficiency for these clinical samples, the fluid that entered and exited the $^{EV}$HB-Chip was also collected and ultracentrifuged to isolate the EVs. For the patient samples whose EGFRvIII status was known ($n = 6$), samples were analyzed in duplicate for the presence of EGFRvIII RNA using digital droplet PCR (ddPCR). First, we quantified the presence of EGFRvIII signal derived from the EVs captured on the $^{EV}$HB-Chip and compared it to the values obtained from ultracentrifugation of the same sample before and after processing with the device (Supplementary Fig. 14). Analyzing similar volumes for each of the conditions, our data showed that the number of positive droplets, or EGFRvIII events, was significantly less for the ultracentrifuged samples before and after the device. As an example, for patient six, the average number of EGFRvIII positive events for the input was 1, outlet: 3, and the $^{EV}$HB-Chip: 53, $n = 2$ (Supplementary Fig. 14). The housekeeping gene (GAPDH) was used to determine the degree of non-specific background RNA message present in the samples (inlet, outlet, and $^{EV}$HB-Chip), with a similar comparison demonstrating markedly higher GAPDH RNA positive events in the ultracentrifuged samples (input: 1857; output: 1696; microfluidic chip: 236, $n = 2$), supporting that EVs isolated from the $^{EV}$HB-Chip yield much higher levels of tumor-specific RNA (EGFRvIII), while having significantly less background RNA from not-tumor EVs (GAPDH).

Using the GBM patient samples, the $^{EV}$HB-Chip demonstrated high specificity for the EGFRvIII mutation in our patients as zero levels were found in our age-matched healthy donors (Fig. 3a–d). While the number of events positive for the EGFRvIII varied across patients, five out six patients exhibited EGFRvIII mRNA transcripts. Additionally, we compared total wild-type EGFR signal levels by ddPCR, and data indicated that more RNA copies were captured on the chip in most patients as compared to healthy donors (Fig. 3b, d). When comparing against the traditional biopsy methods for EGFRvIII detection, our results matched at least one of the biopsy methods (CSF or surgical biopsy) for three patients; with the other three patients being positive for EGFRvIII only by the microfluidic device (Supplementary Table 1).

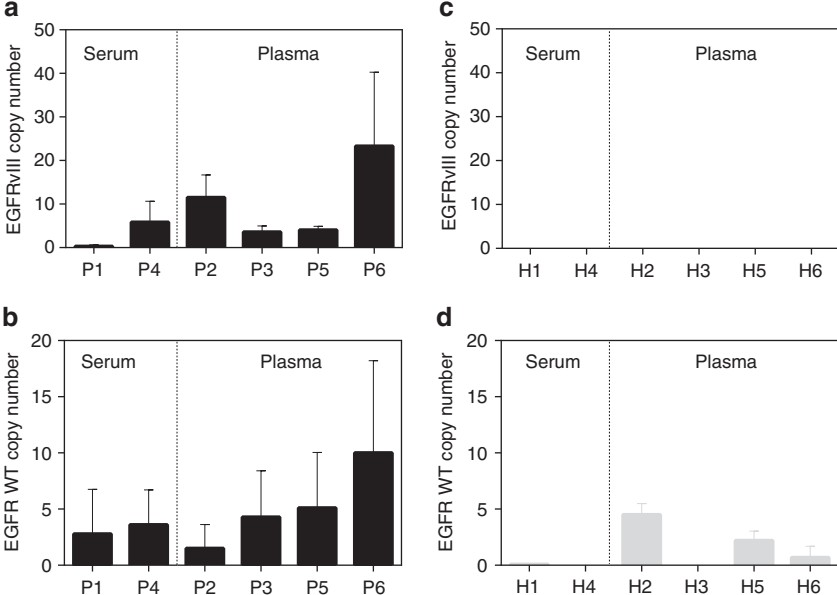

**Fig. 3** Quantification of EGFRvIII mRNA in serum and plasma-derived extracellular vesicles using droplet digital PCR. EGFRvIII and EGFR WT mRNA levels were quantified from six GBM patients (**a**) and (**b**) respectively. Tumor-derived EVs were isolated using the microfluidic platform coated with a cocktail of antibodies, EVs were lysed, and RNA extracted directly from the chip. Droplet digital PCR was used to quantify EGFRvIII, EGFR WT and levels were normalized to total sample input. In (**c**) and (**d**) similar analysis was conducted for six healthy donors (H). Values are expressed as absolute copy numbers of EGFRvIII and EGFR WT mRNA ($n = 3$ technical replicates; ±s.e.m.)

For all patient samples, we analyzed the EVs captured on the ᴱⱽHB-Chip for the presence of characteristic GBM expression signatures by using an amplified RNA sequencing protocol designed for minute quantities of material. Unsupervised clustering analysis of the top one hundred most variant genes showed distinct cluster separation between patients and healthy donors (Fig. 4a, plasma samples ($n = 11$ patients and 4 controls); Fig. 4b, serum samples ($n = 2$ patients and 2 controls)). Many of our patient samples were obtained prior to the initial tumor biopsy, when the official tumor classification was unknown. Upon analysis of the biopsy specimen, patient 11 was determined to be an anaplastic oligodendroglioma (AO). While our sample number is too low to draw any conclusions, it is interesting to note that this AO sample clusters separately from our glioblastoma patients and healthy donors. Lastly, differential expression between EV isolated using ultracentrifugation and the ᴱⱽHB-Chip demonstrated marked changes, indicating that these two methods result in the isolation of distinct populations of EVs (Supplementary Fig. 15).

We have performed the first comprehensive characterization of GBM EV RNA. For the six GBM samples that were analyzed for their EGFRvIII mutational status, a total of 54 GBM genes from a database of primary tumors were detected in tumor-EV transcriptomes (Fig. 4c, Supplementary Table 3). We identified genes previously associated with patient survival (e.g., MAST3, LRRTM2, PEX5L, and GADD45A), disease progression (e.g., ACSL4, AMFR, ARHGEF7, BASP1, EHMT2, MAP3K1, MLLT1, CD151, CDC14B, and E2F3), tumor resistance to radiotherapy or chemotherapy (e.g., ABBC3, PTPRC, ACTN1, EI24, and LCN2), and genes related to stem cell function and putative tumor evolution from a primary or secondary glioblastoma (e.g., CDKN1A, ID1, and ID3, see Supplementary Information). Notably, commonly mutated genes for GBM were found overrepresented and were grouped as signaling genes (e.g., KRAS, NUCB1, PIK3CA, and PRKAR1B, see Supplementary Information). Moreover, gap junction protein and angiogenesis genes were also identified (gap and tight junctions: GJC1, CLDN5; angiogenesis: CXCL5, GUCY1A3, GUCY1B3, see Supplementary Information). Also, we identified 38 cancer-associated genes that were not previously reported in EVs from GBM patients (Supplementary Fig. 16, Supplementary Table 4). Finally, using available databases of genes, we compiled sets of genes uniquely present in each of the four characteristic GBM subtypes[36]. Then, we performed an unsupervised cluster analysis for these gene signatures of EVs isolated from the microfluidic device for patients and healthy donors (Fig. 4d; Supplementary Fig. 17). We found more than 40 genes per subtype that have at least two of their respective landmark genes[36]. For classical subtype: PDGFA, EGFR, and AKT2, for neural subtype: FBXO3, GABRB2, and MBP; for proneural subtype: SOX2 and ERBB3; for mesenchymal subtype: TLR4, RELB, PTPRC, and CASP1/4/8. These results demonstrate that the ᴱⱽHB-Chip captured tumor EVs containing GBM enriched mRNA signatures and potentially reveal transcriptional heterogeneity in GBM tumors.

## Discussion

The use of EVs as biomarkers in cancer research has yet to reach its full potential due to limitations of available approaches that can distinguish between EVs produced by normal and tumor cells[37]. Different microtechnologies have emerged for specific EV isolation[38,39], although many of them operate at very low throughputs[10,40], and are not compatible with the large volumes of serum or plasma (3–5 mL) required for detection of low abundance molecular signatures of mRNAs such as EGFRvIII that has 15–20% frequency in GBM patients[41]. Moreover, current

approaches take advantage of anti-tetraspanins markers for enrichment of EVs, thus producing undetermined and low tumor-EV enrichment ratios[12–14]. Our study is the first to address these key challenges on the EV field with the development of the ᴱⱽHB-Chip, a high-throughput platform that integrates a 3D herringbone microfluidic chaotic mixer with an immunoaffinity nanostructured substrate. We optimized a cocktail of antibodies directed against membrane proteins that are highly expressed on GBM cells that enables the capture of tumor EVs from most patients with glioblastoma. Notice that in our in vitro data using fluorescent tumor EVs, the parental cell line used to generate the EVs (Gli36) has overamplification of EGFRwt and EGFRvIII. Thus, it is not a surprise that when Cetuximab was used as a capture antibody (high binding affinity for both EGFRwt and EGFRvIII), data indicated high capture of tumor EVs when compared to our cocktail of antibodies. However, based on published analysis of primary glioblastoma tumor samples and clinical information, we anticipate that the level of expression of antigens (e.g., EGFR) on the EV surface will be far more heterogeneous than our EVs generated from cell cultures. Thus, we believe that our cocktail approach is necessary to maximize capture of tumor EVs from patients. This was demonstrated with experiments using EVs generated from a parental (glioblastoma) cell line that does not have EGFR amplification (GBM20/3 cells with GFP). In vitro "spike-in" experiments using these EVs shows better performance when our cocktail of antibodies is used vs. Cetuximab alone (Supplementary Fig. 2). Moreover, our design results in higher tumor-EV enrichment ratios when compared to gold-standard methods (>10-fold)[22,23]. This is achieved in part by our unique approach that optimized the PEG-linker length of immobilized antibodies to maximize capture efficiency of tumor EVs on the device while reducing non-specificity EV capture by the steric shielding of PEG[42]. Our device operates at 59% of capture efficiency of tumor-specific EVs, and a limit of detection of 100 EVs per μL, a value comparable with previously reported microfluidic approaches[10,14].

A novel feature of our platform is the ability to release captured tumor EVs from the surface of the device while preserving their cargo contents. Using this feature of the platform, we were able to recover up to 87% of tumor-specific EVs. Having access to these populations of tumor EVs in clinical samples has enormous potential for the investigation of pre-metastatic niche formation, as recent studies demonstrated that EVs from cancer cells from a primary tumor, can remotely prepare distant sites for the spread of tumors in an organ-specific manner[4,43]. Moreover, once isolated and released, these patient-derived EVs can be studied to determine their oncogenic transfer potential to other cells, as it has been revealed that tumor EVs are capable of inducing phenotype changes in surrounding cells[3,44,45]. Additionally, our system does not have any size bias for the captured EVs and being able to isolate exosomes, microvesicles, and oncosomes could have significant biological implications with different cargo packaged in EVs based on their mode of biogenesis[46,47]. Finally, with the characterization of our EV model system, we have shown the imaging benefits provided by the ᴱⱽHB-Chip device that can be correlated to total RNA yield from EVs, thus facilitating the process of optimization for other types of cancer.

One of the hallmarks of glioma is their large spatial and temporal heterogeneity[36,48,49], making the disease difficult to diagnose and to identify genetic mutations. For example, EGFRvIII mutation is one key driver genetic mutation of the disease present in 20% of glioblastoma (GBM) patients[41], but even in positive tumors, not all cells have EGFRvIII[35]. Its identification is important for appropriate diagnosis and treatment. However, tissue or CSF clinical biopsies may not provide

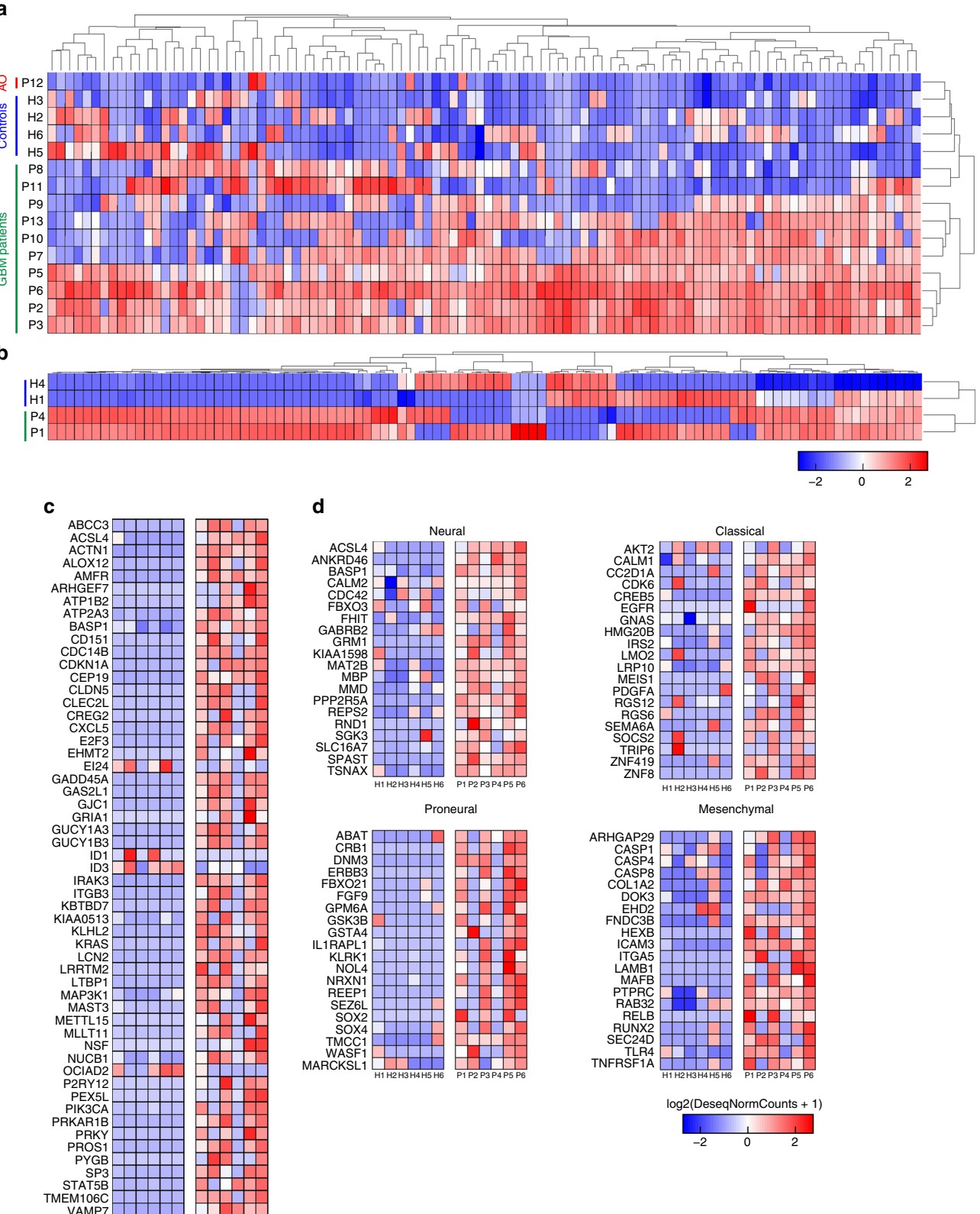

**Fig. 4** RNA sequencing of EVs isolated using the $^{EV}$HB-Chip. Unsupervised clustering of the 100 most variant genes identified in the EVs of patients and healthy donors using the $^{EV}$HB-Chip. For plasma (**a**), data show 10 GBM patients clustering together with one patient clustering outside the group, Patient 12, which was later determined to be an anaplastic oligodendroglioma tumor (AO). For serum (**b**) GBM patients clustered together and separated from healthy donors. **c** Expression heatmaps of log2 normalized gene counts of 54 annotated enriched GBM genes and **d** 20 genes of each GBM subtype classification in patients compared to healthy donors (n = 6 healthy donors (H), n = 13 patients (P))

conclusive results and a non-invasive method to analyze cancer-specific alterations would be favorable. Our small cohort of patients showed that there could be wide discordance between these two methods. It has been shown that EVs carry genetic information of their tumor cell of origin, and in particular can carry EGFRvIII mutant mRNA[3]. The $^{EV}$HB-Chip successfully isolated EVs carrying the EGFRvIII mutation in patient samples and not in healthy donors by using ddPCR. However, we had a higher hit rate for EGFRvIII (5 out of 6) in comparison to the original analysis in the clinic using tumor biopsies and CSF (3 out of 6), suggesting both the possible heterogeneity of the tumors and the influence of the biopsy site. Moreover, our results show a different level of expression of EGFR at the tumor EV RNA level between patients and healthy donors, similar to previous reports using protein analysis[27].

Recently, characterization of the genome and RNA transcriptome of GBM tumors in patients revealed complex cellular ecosystems composed of cells of different phenotypes, and epigenetic landscapes[48,50,51]. On the other hand, GBM-specific EV diversity has not been comprehensively analyzed yet because of the lack of approaches that would allow enrichment of EVs derived from GBM tumors in biofluids. RNA sequencing analysis demonstrated the enrichment capabilities of our device: first, we obtained different gene profiles between the $^{EV}$HB-Chip and ultracentrifugation (Supplementary Fig. 17). From the upregulated genes for the device, we noticed that many of these genes were cancer associated. Second, comparison with databases of known GBM genes showed 54 genes that have the potential to be used as gene panels for disease monitoring. Previously, GBM tumor subtypes were characterized based on differential gene expression into neural, proneural, mesenchymal, and classical subtypes[36]. Each of these subtypes has a specific subset of genes that may help guide brain tumor treatment[36]. However, a recent report of single-cell analysis of solid GBM tumors suggests that individual tumors have heterogeneous mixtures of cells that correspond to different glioblastoma subtypes and hybrid cellular states[48]. Our results showing EVs from different patients with different characteristic subtype genes support those findings and indicating that the heterogeneity of the disease can be potentially recapitulated with RNA sequencing of EVs.

In conclusion, the $^{EV}$HB-Chip is a promising platform that overcomes some of the key limitations in EV isolation, enabling access to unique populations of tumor-specific EVs for downstream molecular characterization with high tumor EV specificity. Applying GBM patient serum/plasma samples to the $^{EV}$HB-Chip identified relatively rare EGFRvIII transcripts, as well as genes specific to GBM subtypes. We believe that the $^{EV}$HB-Chip has strong potential to contribute to the development of EV-based diagnostics and monitoring tools for brain and other types of cancer.

## Methods

**Gelatin functionalization**. Type B gelatin from bovine (bloom 225, Sigma-Aldrich, St. Louis, MO) was dissolved in ddH$_2$0 at a concentration of 4% (w/v) for 1 h. Sulfo-NHS-biotin (Thermo Fisher Scientific, Waltham, MA) was added at a mass ratio of 3.5/1 under stirring conditions. Sodium hydroxide (NaOH) was used to adjust the final pH to 7.4, and the reaction was allowed to proceed overnight at RT. Biotinylated gelatin was dialyzed against ddH$_2$O for 48 h at a 1:1000 ratio (v/v) using a 10 K Mw cutoff dialysis cassette (Thermo Fisher Scientific). Water was changed every 12 h. After dialysis, the solution was freeze-dried for 1 week and storage at 4 °C until use.

**Device fabrication**. Microfluidic devices consisted of 8-Channel herringbone structures that were fabricated using standard photolithography on a 4″ Silicon wafer to create a negative impression of the device[25]. A 10:1 mass ratio of a base and curing agent SYLGARD 184A/B (Dow Corning, Auburn, MI) was poured on the wafer and baked overnight in an oven at 80 °C. Cured polydimethylsiloxane (PDMS) was removed from the mold and holes were punched through the PDMS

for functionalization. Then, devices were bonded to glass slides through oxygen plasma activation at 50 mW, 5 cm$^3$ for 30 s (PX-250, March Plasma Systems, Concord, CA). The microfluidic device fabrication was scaled up by plastic micro-injection molding of cyclic olefin copolymer (COC) at thinXXS Microtechnologies (Germany). The device was manufactured in two separate molds; one for the three-dimensional features, and another for the top layer that was subsequently put together by thermoplastic bonding.

**Microfluidic surface modification**. Different strategies were tested for optimal configuration of capture antibodies on the surface of the microfluid device. Initially, a silane-based chemistry was used;[25] briefly, 4% (v/v) solution of 3 mercaptopropyl trimethoxysilane (Gelest, Morrisville, PA) in ethanol was incubated for 1 h at RT. N-y-malemidedobutyryloxy succinimide ester (Pierce Biotechnology, Rockford, IL) at 0.01 μg mL$^{-1}$ was incubated in ethanol for 30 min at RT in the microfluid channels. After washing the device with PBS, neutravidin (Pierce Biotechnology) at 10 μg mL$^{-1}$ was incubated in the device at 4 °C and storage. Secondly, a recently developed nanostructured substrate was also tested for antibody functionalization[20]. A layer-by-layer assembly technique was used to incorporate biotin–gelatin layers on the surface of the microfluidic device. Biotin–gelatin alone was used as the cationic and ionic polyelectrolyte due to its polyampholyte behavior near neutral pH[52]. Additionally, neutravidin was used to crosslink the thin gel through biotin–streptavidin binding. Each layer of biotin-gelatin at 1% (w/v) was flushed directly in the plasma activated microchannels and incubated for 15 min. Any excess of polymer was removed with PBS and a solution of 100 μg mL$^{-1}$ neutravidin was added and incubated for additional 15 min. A configuration of four layers was found optimal for uniform coverage[20]. The thickness of the nanocoating was characterized using a Dektak 150 Surface Profiler (Veeco, Plainview, NY) with a value of ~150 nm. One last layer of 10 nm streptavidin-coated nanoparticles (Sperotech, Lake Forest, IL) was incorporated into the film to create the nanostructured substrate and increase the local surface area of the added antibodies.

**Antibody conjugation**. Different antibodies were used for tumor-specific EV isolation: EGFRvIII (provided by Dr. Darrel Bigner), EGFR (AF231), hPDGFR (MAB1260), Podoplanin (AF3670), EphA2 (AF3035, R&D Systems, Minneapolis, MN), and the Cetuximab (ImClone LLC, Branchburg, NJ). Each antibody was biotinylated with diverse length spacers to achieve optimal EV capture efficiency. Initially, a zero-length spacer sulfo-Biotin-NHS (Thermo Fisher Scientific) was used according to the manufacturer protocol[25]. Different Mw poly-ethylene-glycol (PEG) spacer: PEG3, PEG6, PEG1.2K, PEG2.4K, PEG5K, PEG10K, PEG20K were used. Briefly, a 1–2 mg mL$^{-1}$ antibody concentration was buffer exchanged using a commercially available kit (CromaLink, Solulink, CA). Then, 100 μL of the antibody solution was mixed with NHS-PEG-Biotin (Creative PEG Works, Chapel Hill, NC) dissolved in 100% DMF. The reaction was allowed to proceed for 2 h at RT. Optimal antibody/PEG-Biotin ratios were calculated according to www.solulink.com. After incubation, biotinylated antibodies were cleaned using a 7K MWCO Zeba Column (Solulink, San Diego, CA) and storage at −80 °C. The biotinylation process was verified using a commercially available UV probe (Solulink) for the low Mw PEG3 spacer; a HABA assay (Thermo Fisher Scientific) was used for all the other conditions. Biotinylated antibodies were incubated in the microfluidic chip for 1 h at 10, 20, or 100 μg mL$^{-1}$ in PBS containing 1% bovine serum albumin (BSA, Sigma-Aldrich).

**Cell culture**. GBM20/3 cell line was generated in the Breakefield laboratory at Massachusetts General Hospital[2], and Gli36 cell line was provided by Dr. Anthony Capanogni from UCLA, Los Angeles. GBM20/3 and Gli36wt and stably infected EGFRvIII (Gli36-EGFRvIII) glioma cells were cultured in DMEM (Invitrogen, Thermo Fisher Scientific) with 10% fetal bovine serum (FBS, Sigma-Aldrich) and 1% penicillin and streptomycin (P/S, Cellgro, Manassas, VA). All cells lines used were passaged using 0.25% trypsin/EDTA (Invitrogen, Thermo Fisher Scientific). Tumor cells were negative for *Mycoplasma* as routinely tested by an enzymatic assay (Promega, Madison, WI).

**EVs production and spike preparation**. To generate fluorescent EVs, Gli36wt and Gli36-EGFRvIII glioma cells were stably infected with PalmtdTomato and PalmGFP[28], respectively, followed by fluorescent activated cell sorting using a BD FACSAria II Cell Sorter. The cells were then cultured in DMEM containing 5% EV-depleted FBS (prepared by ultracentrifugation at 100,000 × g for 16 h to deplete bovine serum EVs) for 48 h. The conditioned medium was centrifuged for 10 min at 300 × g to eliminate cell debris, and the supernatants were centrifuged for 10 min at 2000 × g and filtered through a 0.8 μm filter. Then EVs were pelleted by ultracentrifugation (Optima L-90K Ultracentrifuge, Beckman Coulter, Brea, CA) at 100,000 × g for 70 min. Isolated EVs were resuspended in double 0.22 μm filtered PBS, quantified in size and number (see below for EV quantification) and spiked in serum or plasma from healthy individuals for testing the specificity of the microfluidic device. A 1:1 dilution of serum or plasma in PBS was prepared before running the device. A minimum of 2 mL of solution was used for every sample.

**EV quantification**. Isolated EVs were quantified using a tunable resistive pulse sensing (TRPS) qNano instrument (Izon Science, New Zealand) was used. Different tunable pore size membranes (NP200, NP300, NP400, and NP800) allowed the characterization of size multimodal EV distributions. Briefly, top and bottom fluid cell of the instrument were primed with PBS, and then appropriate calibration beads were forced to flow through the nanopore at pressures between 5 and 15 mbar by a water-based variable pressure module (Izon Science). A similar procedure was applied to EV samples. Acquired data were analyzed using a Control Suite Software provided by the same manufacturer. For the characterization of captured EVs, two methods of release were used. First, proteinase K was used to shave the EVs from the surface of the device and following their recovery with applied flow. Second, a temperature gradient was applied to the surface of the device to disassemble the gelation nanocoating and release the EVs in solution.

**Microfluidic isolation of EVs**. Two or five chips in series were run, depending on the type of samples. For spike EV or patient samples, two or five chips were used, respectively. The first chip was always blank with no functionalization on it. The first chip was used to deplete EVs from platelets. Serum or plasma was pneumatically driven through the chip at a flow rate of 1 mL h$^{-1}$ for 2 h. Next, the microfluidic chip was washed with 2.5 mL of PBS at 2.5 mL h$^{-1}$ to remove any nonspecifically bounded EVs.

**Isolation of EVs with magnetic beads**. Biotinylated Cetuximab with PEG linkers was immobilized with streptavidin-coated magnetic particles (3 µm, Sperotech) for 1 h. Conjugated particles were incubated with spiked Gli36-EGFRvIII EVs in plasma for 2 h. Captured EVs were pull down with a magnet, and gently resuspended in 100 µL PBS for downstream analysis.

**EV confocal imaging**. The COC microfluidic device allowed direct imaging of captured EVs. Micrographs were captured with an LSM510 confocal microscope (Zeiss, Peabody, MA) equipped with a ×63 Zeiss Plan-APOCHROMAT oil objective. Images were collected at the top plane of the device (top plane of the herringbone grooves). A total number of 100 images (10 by 10 images in x- or y-axis) were acquired. Same imaging parameters were used between samples to allow subsequent analyses. Images were processed using Zeiss microscope ZEN software. Semi-quantification of the captured EVs consisted of determining constant threshold of fluorescent intensity between the signal from EVs and noise and automatically calculated using ImageJ[53].

**RNA isolation and quantitative RT-PCR**. Isolated vesicles were lysed inside the chip by pushing 700 µL Qiazol through the device. Subsequently, RNA was extracted from lysates using the miRNeasykit (Qiagen, Valencia, MA) according to the manufacturer's protocol. RNA was eluted from the columns in 50 µL water and concentrated by ethanol precipitation. Next, the RNA quality was assessed using a 2100 Bioanalyzer (Agilent Technologies, Santa Clara, CA) with an RNA 6000 Pico Chip kit (Agilent Technologies). Similar amounts of RNA were reverse transcribed using VILO super kit (Invitrogen, Carlsbad, CA). The relative levels of GFP, TdT, PPBP, EGFR, EGFRvIII, and GAPDH were assayed by single tube TaqMan assays (Life Technologies, Carlsbad, CA): GFP, Mr00660654_cn; TdTomato, AI39R57; PPBP, Hs00234077_m1; GAPDH, Hs02758991_g1 [EGFRvIII].

**Digital PCR**. RNA was reverse transcribed into 20 µl cDNA reactions using the Sensiscript Reverse Transcription Kit (Qiagen) according to the manufacturer's instructions. A volume of 5 µl of cDNA was used as input in duplicate reactions for each assay (EGFRvIII and EGFRwt). EGFRvIII primers were as follows: EGFRvIII Forw: CTGCTGGCTGCGCTCTG; EGFRvIII Rev: GTGATCTGTCACCACA-TAATTACCTTTC; EGFR probe: TTCCTCCAGAGCCCGACT; EGFRwt primers were as follows: EGFR Forw: TATGTCCTCATTGCCCTCAACA; EGFR Rev: CTGATGATCTGCAGGTTTTCCA; EGFR Probe: AAGGAATTCGCTCCACTG. About 20,000 droplets were generated using the AutoDroplet generator (Bio-Rad, Hercules, CA). PCR conditions were as follows: 95 °C for 10 min, 39 cycles at 94 °C for 30 s and 61 °C for 1 min. The last stage was 98 °C for 10 min followed by 4 °C. Droplets were analyzed using the Droplet Reader (Bio-Rad). Gates were set to exclude all events from the cDNA no-template control sample. All events above the no-template control gates were considered positive. Concentrations were calculated in auto mode using the Bio-Rad software. The patient samples were run without knowing a priori which results from tissue or CSF biopsy were positive for EGFRvIII.

**Library preparation for RNA sequencing**. EVs were lysed with 700 µL of qiazol and RNA was extracted using a mRNAeasy kit form Qiagen (Hilden, Germany). RNA was amplified using a modified protocol[54] and sequenced at the Broad Institute of Harvard and MIT. In short, amplified cDNA was synthesized from the entire cell lysate using the SMARTer Ultra Low Input RNA Kit for Sequencing—v3 kit (Clontech Laboratories). PCR amplification following second strand synthesis was run for 18 cycles. One nanogram of amplified cDNA was loaded into the Nextera XT kit. Normalization was done using the KAPA SYBR® FAST Universal qPCR Kit (Kapa Biosystems) rather than the bead-based normalization in the Nextera XT kit. The pooled libraries were sequenced on multiple lanes of a HiSeq2000.

**RNA sequencing analysis**. The initial quality control of the data was carried out using the tool FASTQC[55]. Once it was determined that the samples were good, the alignment of the samples to the reference genome was conducted using STAR aligner[56]. The duplicate reads were marked using PICARD[57]. and removed using SAMTOOLS[58]. The resulting BAM files were used to quantify the read counts per gene using Htseq-count program[59]. The downstream analysis was carried out in R statistical programming language[60]. To get an insight into the data, we selected 100, 500,1000, and 2000 most variant genes and did a hierarchical clustering for all samples based on the expression of these genes. We used the DESEQ2 package[61] in R for the differential expression analysis between the two clusters and to finally get a list of differentially expressed genes between any two conditions of interest. The heatmaps were plotted using the heatmap.2 function in gplots package in R.

**Statistical analysis**. Data are expressed as mean ± s.e.m. Significant differences between mean values were evaluated using Student's t-test, one-way ANOVA.

**Clinical samples**. Blood samples from healthy donors and cancer patients were collected with informed consent according to an institutional review board (IRB) protocol 05–300, at the Massachusetts General Hospital. A total of 13 brain cancer patients and 6 healthy donors were included in this study. A volume of 10 mL blood samples were collected by venipuncture into a BD Vacutainer SST tube (#367985) or BD Vacutainer PPT (#362788) for serum and plasma, respectively. Samples were left to clot for 30 min at RT and processed according to the manufacturer's protocol within 2 h of collection. Serum or plasma was filtered through a 0.8 µm filter and run through the microfluidic device or stored at −80 °C for later processing.

**Data availability**. The data that support this study are available from the corresponding author. Raw RNA-sequencing data with annotations is available at under accession code GSE106804.

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

## Acknowledgements

We acknowledge all the patients and healthy volunteers who participated in this study. We acknowledge Wooseok Kim for device functionalization, Erik R. Abels for technical help with RNA extraction and Octavio Hurtado for technical expertise. We thank Dr. Darrel Bigner from Duke University for providing the EGFRvIII antibody. We thank Dr. Anthony Capanogni from UCLA for providing the Gli36 cell line. This work was supported by NIH grants CA069246 (X.O.B.), U19CA179563 (supported by the NIH Common Fund, through the Office of Strategic Coordination/Office of the NIH Director; X.O.B.), Voices Against Brain Cancer (X.O.B. and C.P.L.), the National Institute for Biomedical Imaging and Bioengineering (NIBIB) EB008047 (M.T.), NIH P41 EB002503-11 (M.T.)., and the Wang Pediatric Brain Tumor Collaborative (S.L.S.); C.P.L. was supported by the Canadian Institute of Health Research (CIHR).

## Author contributions

E.R. wrote the manuscript with input from all authors. E.R. and S.L.S. designed the microfluidic platform. E.R., K.E.V., C.P.L., N.A.A., F.P.F., A.H.K. and S.L.S. designed and performed experiments. E.R., K.E.V., C.P.L, X.O.B. and S.L.S. analyzed the data. E.R., K.E.V., C.P.L., M.T., X.O.B. and S.L.S. discussed research directions. M.Z. and L.B. performed RNA extraction, ddPCR experiments, and analyzed the ddPCR data. B.A., V.T. and D.T.T. performed bioinformatics analysis. B.A. performed gene classification analysis. F.H.H., L.V.S., B.V.N. and B.S.C. provided clinical samples and patient clinical information.

## Additional information

**Competing interests:** The authors declare no competing financial interests.

