## [Peer Review File · Nature Communications]

Reviewers' comments:

Reviewer #1, an expert in microfluidic devices (Remarks to the Author):

This paper by Prof. Shannon L. Stott presents microfluidic chip technology for capture and downstream analysis of exosomes or extracellular vesicles from GBM patient plasma and serum. This is a nice paper and should be published. However, i have the following comments:

The throughput and flow rate is a concern and some value of throughput or time of analysis should be mentioned right in the abstract. take that head on.

Also, maybe i missed this in the details but how do we make sure that the cfDNA is not analyzed in the measurements?

Also, controls with just capture without the herringbones would have been nice. its not clear to me that particles of these small sizes would be affected by the flow and what the shear stress are that would maximize capture. The particles are getting to the capture points mainly by diffusion i think.

Also, is there a way to state capture efficiency ? how many EVs are being captured ? i realize the control measurements are from current/older methods.

Figure 2g: what does it mean when the bar is labeled serum or magnetic beads. Did the authors do it themselves ? i am not sure if this comparison is good as the magnetic bead capture can be optimized i am sure and can be higher. maybe some comments can be made around this. of course if you compare the through put and time of analysis versus the magnetic bead capture, its not clear how much better the current technique is.

Reviewer #2, an expert in identification of tumour cells using micro-devices (Remarks to the Author):

This manuscript describes a microfluidic platform for tumor-derived isolation of extracellular vesicles (EVs) using immunoaffinity-based capture. The authors tested different antibody immobilization strategies to improve capture of EVs and the flow rate was optimized to specifically isolate tumor-derived EVs. The incorporation of thermo-sensitive gelatin nanocoating allowed release of captured exosomes for downstream analysis. The device was compared against standard methods of EV isolation (ultracentrifugation and magnetic bead separation) and this device yields a 10-fold higher mRNA, a significant achievement. A small cohort of GBM patient samples (6 patient) was used to further validate performance and EVs were isolated from serum or plasma. The isolated EVs were further analyzed using RNA sequencing analysis, and EGFRVIII mutant mRNA was detected in the isolated EVs from patient samples; the number of patient samples is quite small, but this is a good proof-of-concept data set.

Overall the manuscript is well written and clear. The herringbone microfluidic chip has been used in other studies, but here the authors incorporated a nanostructured coating and a PEG linker for the efficient capture of exosomes which is a novel approach.

Several issues could be addressed to improve the manuscript:

1- Figure 1f shows that the use of cocktail antibody against EGFR, EGFRvIII, BAF231, podoplanin and PDGFR results in less capture of EVs compared with Cetuximab. Authors should include an explanation regarding this observation.

2- In order to increase the capture of tumor-specific EVs, EVs were first processed using a chip

coated with control IgG antibody. The authors should clarify that the EVs are not non-specifically captured in the first chip. This can be done by extracting the captured EVs from IgG coated device and performing RT-qPCR.

3- As mentioned in the result section, the approach is not universal and it requires the identification of tumor specific antigens present on the EV surface. This should be clarified in the discussion section.

4- In the main text or figure caption, authors should better explain how they counted the captured EVs.

5- The authors cited ref 11 because of the use of microfluidic approach for EV-isolation; however, this paper used ultracentrifugation for exosome isolation.

6- A reference should be provided for the following sentence (page 3):
"or use anti-EpCAM antibodies that are also expressed on normal epithelial cells, thereby limiting the specificity of the isolated EVs."

7- Figure 2f, 2g and 2h do not reflect the text and should be checked

8- Figure 2g does not show the result of ultracentrifugation isolation. Authors should include this result for better assessment.

9- Figure 2 SI: dilution ratio should be shown in the graph.

Reviewer #3, an expert in RNA-seq of cancer cells (Remarks to the Author):

Clinical care of glioblastoma multiforme (GBM) patient is challenging due to the limited use of biomarkers. Authors provided new method to monitor the clinical progression of GBM patients through sensitive and specific exosome isolation and RNA sequencing. Basically they modified herringbone circulating tumor cell (CTC) chip for the isolation of exosome vesicle and analyzed RNA profile. In EVHB-Chip, they optimized the microfluid device for exosome isolation through testing fluidic control methods and various linkers for antibody conjugation. Finally authors maximized the efficiency and the specificity of exosome vesicle capture in EVHB-Chip.

1. To show the performances, authors compared the ultracentrifugation methods (Fig. 2b), and magnetic beads (Fig. 2g). However, it is not clearly described and analyzed in parallel with exosome isolation methods to show the performances of EVHB-Chip?

2. Is it possible to calculate the limit of detection on exosome vesicle?

3. Authors showed six positive GBM cases to demonstrate the analytical performances of EVHB-Chip, but the number of patients is not enough to draw a conclusion.

4. If the expression profile of exosome vesicle contained the features for four subtypes, authors can classify the patients into subtypes based on their exosome vesicle expression patterns.

5. Can you explain why Cetuximab only is better than cocktail antibody to capture the exosome vesicle?

Response Manuscript NCOMMS-17-07175

R. We would like to thank all three reviewers for their careful reading of our manuscript and for their thoughtful comments and questions. In response to their critiques, we have completed a series of new experiments ($n = 8$; 3 replicates each), including adding additional patient samples (7 additional patients). We believe that this new data strengthens our manuscript and helps to better characterize our technology.

Reviewer #1, an expert in microfluidic devices (Remarks to the Author):

This paper by Prof. Shannon L. Stott presents microfluidic chip technology for capture and downstream analysis of exosomes or extracellular vesicles from GBM patient plasma and serum. This is a nice paper and should be published. However, i have the following comments:

1. The throughput and flow rate is a concern and some value of throughput or time of analysis should be mentioned right in the abstract. take that head on.

R. We agree with the reviewer that the time of analysis is of importance for the feasibility of using extracellular vesicles as a diagnostic tool in the clinic. Our microfluidic platform allows for a rapid and easy protocol that results in tumor-specific EV-RNA isolation from serum or plasma within 3 hours. We have added this information to the abstract and also added a few sentences to explain the total time of our protocol in the results section on page 10.

2. Also, maybe i missed this in the details but how do we make sure that the cfDNA is not analyzed in the measurements?

R. We used multiple approaches to ensure that we only analyzed EV-RNA in our experiments and not cfDNA. Prior to lysing the EVs (when they were still captured on-chip), wash steps are conducted to remove any non-specifically bound proteins and nucleotides from inside the chip. Following lysis of the EVs, RNA was isolated using the Qiagen miRNeasy Mini Kit, which purifies total RNA including RNA from approximately 18 nucleotides (nt). This kit is designed to remove most DNA from the lysates. Subsequently, we analyzed the enrichment for tumor-specific EV-RNA with TaqMan probes that were designed to span introns and therefore do not amplify DNA.

In addition to using care to make sure cfDNA is not being analyzed, we performed additional experiments to show that non-specificity is minimal with our approach. As our baseline, we used RT-PCR to quantify the total amount of PPBP (pro-platelet basic protein) present in EVs isolated using ultracentrifugation of plasma ('input' in Supplementary Fig 3a). Similar to GAPDH, we then use PPBP as a marker of non-specific RNA signal for our EVs. Using the same methods, we then quantify the amount of PPBP signal retained on the device, using different capture antibodies ("Cocktail" and "Cetuximab" in Supplementary Fig. 3a). Our result shows minimal PPBP message is present on the chip (coming from EVs of non-tumor origin) relative to EVs isolated from bulk plasma ("input"). The following graph was added as Supplementary Figure 3a.

Supplementary Figure 3a. Comparison of retention of non-specific EVs on the ^{EV} HB-Chip by RT-PCR quantification of the pro-platelet basic protein (PPBP). The total amount of PPBP present in EVs isolated using ultracentrifugation of bulk plasma ('input') is compared to the signal present in EVs isolated on the device. The same cocktail of antibodies used in **Supplementary Figure 2** was used for these experiments. Cetuximab was used at a concentration of 20 $\mu\text{g/mL}$.

3. Also, controls with just capture without the herringbones would have been nice. its not clear to me that particles of these small sizes would be affected by the flow and what the shear stress are that would maximize capture. The particles are getting to the capture points mainly by diffusion i think.

R. We thank the reviewer for raising this question. To demonstrate the additional benefit of the herringbone structures (and resulting chaotic mixing of the plasma), we performed additional experiments in which we compared the amount of EVs captured on a 3D herringbone structure versus a flat channel. This new data shows that

incorporating herringbone structures results in ~60 % more EVs captured. The following figures have been added as supplemental material and are described in the manuscript text.

Supplementary Figure 12a. Average fluorescence intensity measurement of EVs captured on a microfluidic flat channel and an ^{EV}HB-Chip. Td-tomato fluorescence EVs were spiked in PBS at 50 million EVs/mL (n = 3 per experimental group).

The reviewer is also right to point out that there are microfluidic platforms that take advantage of a “diffusion driven force” to increase nanoparticle separation efficiency.¹ However, for our approach, we utilize the combined effect of (1) the anisotropic flow / mixing from 3D herringbone structure (**Supplementary Figure 12a**) and (2) the optimized surface chemistry to promote capture of small particles (**Fig. 2c**) to maximize capture of tumor-specific EVs. With this additional experiment that the reviewer has suggested, we believe that the readers can better discern the benefit of each component.

For completeness, we also calculated the impact of shear stress on EV binding inside a microfluidic channel. We used a Hele-Shaw device (**Supplementary Figure 12b**) in which the shear stress along the axis of the chamber decreases linearly along the chamber length without changing the input flow rate.²⁻⁴

Supplementary Figure 12b. Schematic representation of the Hele-Shaw device used to calculate the shear stress of EVs on microfluidic channels.

The shear stress is calculated based on the equation derived by Usami:⁴

$$\tau_w = \frac{6\mu Q}{h^2 w} (1 - x/L)$$

Where: L is the length in mm, w is the width in mm, h is the height in μm (for our case was set to $47 \mu\text{m}$), μ is the viscosity of the fluid carrying the EVs, Q is the input flow rate, and x/L represents the normalized position at which the shear stress will be calculated.

We performed experiments at two different flow rates (Q): At 1 mL/h , the shear stress experienced by the EVs was 0.11 to 1.1 dyn/cm^2 with a drop of 19.6% in the fluorescence intensity of captured EVs. When the flow rate was increased to 3 mL/h , the shear stress values were between 0.32 to 3.17 dyn/cm^2 with a drop of 92.8% in the fluorescence intensity of captured EVs. Therefore, choosing a flow rate of 1 mL/h was appropriate to keep specificity and high EV capture percentages. We have added the above text and the following figure as **Supplementary Figure 12c**.

Supplementary Figure 12c. Characterization of the shear stress in the Hele-Shaw Device for two different flow rates ($n = 3$ per condition). We performed experiments at two different flow rates (Q): At 1 mL/h (blue circles), the shear stress experienced by the EVs was 0.11 to 1.1 dyn/cm² with a drop of 19.6 % in the fluorescence intensity of captured EVs. When the flow rate was 3 mL/h (open circles), the shear stress values were between 0.32 to 3.17 dyn/cm² with a drop of 92.8 % in the fluorescence intensity of captured EVs. Therefore, we chose a flow rate of 1 mL/h to maximize specificity and EV capture.

4. Also, is there a way to state capture efficiency ? how many EVs are being captured ? I realize the control measurements are from current/older methods.

R. Capture efficiency, a standard metric for whole cell isolation, is not as easy to determine for EVs due to their size (difficult to image and count) and high frequency (in the millions). With that said, in this revision, we did our best to estimate the capture efficiency of the ^{EV}HB-Chip. Specifically, the capture efficiency was calculated by spiking a known number of EVs per mL into PBS. EV concentrations between 35 to 50 million of particles per mL of PBS were used for these experiments. The initial concentration of EVs was determined using a qNano particle counter from Izon Science. We performed three separate experiments, in which the number of particles was

analyzed before and after the samples were run through the ^{EV}HB-Chip to establish a ‘mass balance’ of sorts. Our results indicated that the device had a capture efficiency of 58.77 % ± 5.37 (mean ± s.em.). The specific depletion of the EVs on the device can also be visualized by comparing measurements of particle rates (a raw measure of concentration) before and after running the sample through our device. The following figures has been added to supplemental material, and the values of capture efficiency of our device have been added to the manuscript. Also, all the data presented for the control measurements are new.

Supplementary Figure 10. (a) Capture efficiency on the ^{EV}HB-Chip for spiked TdTomato Glio36 EVs in PBS. (n = 3, ^{EV}HB-Chip conditions used were the same as our patient samples (see Methods)). **(b)** Quantification of the depletion of EVs before and after it was run through the ^{EV}HB-Chip. The lower particle count after the sample was flow through the ^{EV}HB-Chip is an indication of EV depletion. The initial and final particle count of EVs was calculated using a qNano instrument before and after the samples were run through the ^{EV}HB-Chip.

5. Figure 2g: what does it mean when the bar is labeled serum or magnetic beads. Did the authors do it themselves ? i am not sure if this comparison is good as the magnetic bead capture can be optimized i am sure and can be higher. maybe some comments can be made around this. of course if you compare the through put and time of analysis versus the magnetic bead capture, its not clear how much better the current technique is.

R. In Fig. 2g, The bar labeled as serum corresponds to EVs isolated from serum using ultracentrifugation. We have changed the label for clarity (see also response to

comment 8 for reviewer 2). For the magnetic bead separation of EVs we used a recent protocol described by Shao et al.⁵ designed for GBM samples (but using our cocktail of antibodies), to compare against our ^{EV}HB-Chip. As indicated in Fig. 2g the value of the current technique resides on the enrichment of the tumor EVs in this case the GFP-EVs from Gli36-EGFRvIII cells.

The reviewer also raises the concern of throughput vs. yield. For this three-way comparison, we selected a volume of plasma to be processed such that the processing times would be equal for the magnetic bead isolation and ^{EV}HB-Chip (250uL of plasma was used). For the protocol followed by magnetic bead isolation of EVs, using the same volume of plasma / same time for processing, we have significantly higher levels of tumor specific RNA using the ^{EV}HB-Chip (Fig. 2g). Similarly, our device has significantly higher enrichment of tumor-specific RNA in comparison to ultracentrifugation, and approach that is often more time intensive and requires expensive equipment. Thus, we believe our data support that our technique is superior to these methods for tumor specific enrichment of EVs, under the conditions tested.

Reviewer #2, an expert in identification of tumour cells using micro-devices (Remarks to the authors)

This manuscript describes a microfluidic platform for tumor-derived isolation of extracellular vesicles (EVs) using immunoaffinity-based capture. The authors tested different antibody immobilization strategies to improve capture of EVs and the flow rate was optimized to specifically isolate tumor-derived EVs. The incorporation of thermo-sensitive gelatin nanocoating allowed release of captured exosomes for downstream analysis. The device was compared against standard methods of EV isolation (ultracentrifugation and magnetic bead separation) and this device yields a 10-fold higher mRNA, a significant achievement. A small cohort of GBM patient samples (6 patient) was used to further validate performance and EVs were isolated from serum or plasma. The isolated EVs were further analyzed using RNA sequencing analysis, and EGFRvIII mutant mRNA was detected in the isolated EVs from patient samples; the number of patient samples is quite small, but this is a good proof-of-concept data set. Overall the manuscript is well written and clear. The herringbone microfluidic chip has been used in other studies, but here the authors incorporated a nanostructured coating and a PEG linker for the efficient capture of exosomes which is a novel approach. Several issues could be addressed to improve the manuscript:

1- Figure 1f shows that the use of cocktail antibody against EGFR, EGFRvIII, BAF231, podoplanin and PDGFR results in less capture of EVs compared with Cetuximab.

Authors should include an explanation regarding this observation.

R. We thank the reviewer for raising this point, as we failed to clearly explain our results or place them in the proper context (a similar concern was raised by Reviewer 3). For Figure 1f, our *in vitro* data using fluorescent tumor EVs, the parental cell line used to generate the EVs (Gli36) has over amplification of EGFRwt and EGFRvIII. Thus, it is not a surprise that when Cetuximab was used as a capture antibody (high binding affinity for both EGFRwt and EGFRvIII), data indicated high capture of tumor EVs. For our 'single antibody experiments,' we used 20ug/mL of the antibody for capture. When the antibody was included as part of a cocktail of antibodies, the amount of each antibody (in this case Cetuximab) 'on-chip' was decreased to 10ug/mL to accommodate the other antibodies. Thus, we believe the lower capture of these EGFR amplified EVs with the cocktail is simply an artifact of less Cetuximab being present in the cocktail (10ug/mL in the cocktail vs. 20ug/mL in the "Cetuximab only" chip).

Based on published analysis of primary glioblastoma tumor samples and clinical information, we anticipate that the level of expression of antigens on the EV surface will be far more heterogeneous than our EVs generated from cell cultures. Thus, we believe that our cocktail approach is necessary to maximize capture of tumor EVs from patients. While EGFR overexpression and EGFRvIII will be present in some patient samples, it will also be absent in many others. Thus, we believe that it would be more appropriate to have a cocktail of antibodies that account for tumor heterogeneity. To demonstrate these points better, we have added new data using EVs generated from a parental (glioblastoma) cell line that does not have EGFR amplification (GBM20/3 cells with GFP). *In vitro* 'spike-in' data using these EVs shows better performance when our cocktail of antibodies is used versus Cetuximab alone (Supplementary Figure 2c). We have also included additional data in Supplementary Figure 2 to further demonstrate gains in tumor EV isolation using our cocktail approach versus a single antibody against EGFRvIII (Supplementary Figure. 2a and 2b).

For our patient samples, we increased the concentration of Cetuximab to 100ug/mL on-chip due to its inexpensive cost, while also still including the other antibodies for capture. We have added Supplementary Figure S2 to the manuscript, and we have also adjusted the caption of figures, discussion section and methods to clarify the above points.

Supplementary Figure 2 . (a) Comparative confocal microscopy images of tumor EVs captured on the surface of the ^{EV} HB-Chip with different isolation antibodies (e.g., concentration and different antibodies). **(b)** Normalized quantification of captured EVs on the surface of the ^{EV} HB-Chip. **(c)** Comparison of the tumor -specific EV capture for different antibodies on the surface of the ^{EV} HB-Chip. For data shown in (a) and (b) EVs were obtained from Gli36 cells; data in (c) was obtained using EVs from GBM20/3 with palmGFP. The cocktail of antibodies used for these experiments consisted of EGFRvIII, BAF231, PDG FR, Podoplanin, and Cetuximab; each antibody at a concentration of 10ug/mL.

2- In order to increase the capture of tumor-specific EVs, EVs were first processed using a chip coated with control IgG antibody. The authors should clarify that the EVs are not non-specifically captured in the first chip. This can be done by extracting the captured EVs from IgG coated device and performing RT-qPCR.

R. We thank the reviewer for this suggestion and we have performed additional experiments to show that the IgG chip minimally depletes tumor specific EVs. First, we quantified the changes in EV concentration before and after flowing EVs through the device. We found that less than 5% of the spiked EVs remained on the IgG Chip (**Supplementary Figure 7a**, below). As the reviewer suggested, we also performed PCR to further validate our findings (see **Supplementary Figure 7b**, below). Using ddPCR, we analyzed the specificity of capture by quantification of EGFRvIII copy numbers (using EVs generated from Gli36, an EGFRvIII positive cell line). For spike-in experiments using these EVs, the ^{EV}HB-Chip coated with Cetuximab had 10-fold higher copy numbers of EGFRvIII relative to the IgG chip, further suggesting minimal capture of tumor specific EVs on the IgG chip. The following figure was added as supplemental material and is described in the manuscript.

Supplementary Figure 7. (a) Tumor-EV depletion on the ^{EV}HB-Chip when using an irrelevant, IgG control antibody (gray) and Cetuximab (red). (b) Comparison of the specificity of capture for GFP Gli36 EVs for Cetuximab (red) and an IgG control antibody (gray) on the EVHB-Chip; ddPCR was used to quantify the total number of EGFRvIII copies. All experiments used EVs generated from GFP Gli36 parental cells.

3- As mentioned in the result section, the approach is not universal and it requires the identification of tumor specific antigens present on the EV surface. This should be clarified in the discussion section.

R. The reviewer is correct that for the approach may require adjustment of capture antibodies for each cancer that we wish to explore. For this study, we selected antibodies that were directed against membrane proteins that have been shown to be highly expressed on GBM cells and GBM-EVs. We have added a few sentences to the discussion for clarification and removed the reference to 'universal'.

4- In the main text or figure caption, authors should better explain how they counted the captured EVs.

R. We apologize that this was not clear. Captured EVs on the chips were imaged with an LSM510 confocal microscope (10 images per x- and y-axis; a total of 100 images), and semi-quantified using ImageJ's "Analyze Particle" function by applying the same thresholds across all of the samples. These sentences have been added to the methods section:

EV Confocal Imaging: The COC microfluidic device allowed direct imaging of captured EVs. Micrographs were captured with an LSM510 confocal microscope (Zeiss, Peabody, MA) equipped with a 63x Zeiss Plan-APOCHROMAT oil objective. Images were collected at the top plane of the device (top plane of the herringbone grooves). A total number of 100 images (10 images per x- or y-axis) were acquired. Same imaging parameters were used between samples to allow subsequent analyses. Images were processed using Zeiss microscope ZEN software. Semi-quantification of the captured EVs consisted of determining the constant threshold of fluorescent intensity between the signal from EVs and noise and automatically calculated using ImageJ.⁶

5- The authors cited ref 11 because of the use of microfluidic approach for EV-isolation; however, this paper used ultracentrifugation for exosome isolation.

R. We thank the reviewer for pointing out this error, we have changed the reference to the correct publication we intended to reference here (Chen et al., Microfluidic isolation and transcriptome analysis of serum microvesicles. *Lab Chip*, 2010).⁷

6- A reference should be provided for the following sentence (page 3):
"or use anti-EpCAM antibodies that are also expressed on normal epithelial cells, thereby limiting the specificity of the isolated EVs."

R. We have added a reference to the sentence that describes the expression of EpCAM on epithelia.⁸

7- Figure 2f, 2g and 2h do not reflect the text and should be checked

R. We apologize for the confusion and thank the reviewer for the careful reading of our

work. Figure 2f, 2g, and 2h have been properly described through the text in the manuscript.

8- Figure 2g does not show the result of ultracentrifugation isolation. Authors should include this result for better assessment.

R. We apologize for the confusion on the labeling of Figure 2g. The experiments presented are indeed a comparison between ^{EV}HB-Chip, magnetic nanoparticle separation and ultracentrifugation. We have corrected the text and have also slightly modified our original description of the experiments.

“We tested the final design of the ^{EV}HB-Chip against standard methods of EV isolation. Ultracentrifugation and magnetic bead separation are widely used for the isolation of EVs.⁹ Samples of PalmGFP-EGFRvIII GBM EVs spiked into the plasma were divided in triplicates and run independently on each platform. For ultracentrifugation samples were centrifuged for 2 hrs at 100,000 x g; for magnetic separation, samples were incubated with 3 μm magnetic, antibody coated, polystyrene beads for 2 hrs; and the ^{EV}HB-Chip was run for the same amount of time (see Methods). For all three platforms, isolated EVs were lysed and homogenized with 700 μL of Qiazol buffer. Quantification of the tumor EV-specific message (GFP) indicated that the microfluidic platform had a 10-fold higher GFP mRNA content compared to ultracentrifugation and bead-based separation methods (**Fig. 2g**).”

9- Figure 2 SI: dilution ratio should be shown in the graph.

R. We have added the dilution ratio to the Supplementary Figure 2.

Reviewer #3, an expert in RNA-seq of cancer cells (Remarks to the Author):

Clinical care of glioblastoma multiforme (GBM) patient is challenging due to the limited use of biomarkers. Authors provided new method to monitor the clinical progression of GBM patients through sensitive and specific exosome isolation and RNA sequencing. Basically they modified herringbone circulating tumor cell (CTC) chip for the isolation of exosome vesicle and analyzed RNA profile. In EVHB-Chip, they optimized the microfluid device for exosome isolation through testing fluidic control methods and various linkers for antibody conjugation. Finally authors maximized the efficiency and the specificity of exosome vesicle capture in EVHB-Chip.

1. To show the performances, authors compared the ultracentrifugation methods (Fig.

2b), and magnetic beads (Fig. 2g). However, it is not clearly described and analyzed in parallel with exosome isolation methods to show the performances of ^{EV}HB-Chip?

R. We apologize for the confusion. Fig 2b is a comparison between the ^{EV}HB-Chip with different antibodies and a standard ultracentrifugation method. Conversely, Fig. 2g presents a different set of experiments in which we compare the three different methods: ^{EV}HB-Chip, magnetic bead separation, and ultracentrifugation. In order to clarify this, we have been renamed the category serum as ultracentrifugation and took care to make sure the text was straight forward.

“We tested the final design of the ^{EV}HB-Chip against standard methods of EV isolation. Ultracentrifugation and magnetic bead separation are widely used for the isolation of EVs.⁹ Samples of PalmGFP-EGFRvIII GBM EVs spiked into the plasma were divided in triplicates and run independently on each platform. For ultracentrifugation samples were centrifuged for 2 hrs at 100,000 x g; for magnetic separation, samples were incubated with 3 μm magnetic, antibody coated, polystyrene beads for 2 hrs; and the ^{EV}HB-Chip was run for the same amount of time (see Methods). For all three platforms, isolated EVs were lysed and homogenized with 700 μL of Qiazol buffer. Quantification of the tumor EV-specific message (GFP) indicated that the microfluidic platform had a 10-fold higher GFP mRNA content compared to ultracentrifugation and bead-based separation methods (**Fig. 2g**).”

2. Is it possible to calculate the limit of detection on exosome vesicle?

R. Previously, the limit of the detection of EVs has been calculated by other groups using the average fluorescence intensity of EVs.^{10,11} Following their approach, we prepared a series of dilutions of spiked EVs on PBS from a known stock concentration. Then, the captured EVs were imaged on a fluorescent microscope with the same exposure times for all the different titration conditions. It is important to mention here that we are not measuring individual EV fluorescence; the dense capture nanoparticle layer deposited on the surface of the device aggregates the EVs on the surface. Thus they can be visualized and quantified. The aggregation of EVs has been previously shown to be appropriate to quantify limits of detection.^{10,11} By using this method, we found that our current limit of detection is 100 EVs/μL, far below what is thought to circulate in cancer patient blood and is also comparable with previously reported microfluidic approaches for general EV isolation.^{10,11} The following figure was added as supplemental material and the limit of detection was added to the text.

Supplementary Figure 11. The limit of detection of the EVHB-Chip by average fluorescence intensity of TdTomato Glio36 EVs. ΔF represents the change in fluorescence intensity between a blank device and a device that was run with different concentrations of EVs (Particles/ μL).

3. Authors showed six positive GBM cases to demonstrate the analytical performances of ^{EV}HB-Chip, but the number of patients is not enough to draw a conclusion.

R. We agree with the reviewer that no clinical conclusions should be made from a study that includes six patients. Our work was never intended to be a clinical study, but a demonstration of the clinical utility of our new microfluidic device using patient samples. The scope of this work also includes the development and validation of our tumor-specific EV isolation assay, wherein we discovered that antibody specific capture of EVs requires additional modifications from approaches used for cells. Glioblastoma is not a very common cancer, and patients with the EGFRvIII mutation are even more rare. Thus, we were limited to banked samples for our EGFRvIII analysis. We believe that results from six patients as a proof of concept is on par with the field and also validates our work with EVs cultured in vitro. To address the reviewer's concern that this is still too small of a cohort, we have since run an additional seven samples from patients with brain cancer (6 GBM, 1 anaplastic oligodendroglioma (AO)) and performed RNASeq on

the EVs isolated using our ^{EV}HB-Chip. These patients were treated at the Massachusetts General Hospital and the samples were obtained prior to their initial diagnostic biopsy. Our previous EGFRvIII samples were from frozen plasma that was stored at -80C and obtained through the GBM Consortium led by Dr. Bob Carter (additional samples were not available). Due to these differences in sample acquisition, storage and disease stage, we were careful not to draw too many conclusions from the overall grouping. With the inclusion of the new RNASeq data from the seven patients, we evaluated the top differentially expressed genes across all samples as well as clustering analysis. We were pleased to see that despite differences in institution (UC San Diego vs. MGH), storage (-80C vs. real time) and treatment time point (pre-biopsy vs. advanced disease), all our GBM samples clustered together and all of our healthy donors clustered together (new Figure 4a and 4b). We noticed that one patient (Patient 12) was not clustering with our patients, nor was it clustering with our healthy donors. Dr. Brian Nahed, the neurosurgeon that obtained these samples then informed us that this patient was originally thought to have a GBM tumor pre-biopsy, but after surgery, pathology determined that their tumor was an anaplastic oligodendroglioma (AO). We are excited to see that our approach was able to identify differences between these two types of brain tumors, all from less than 2mLs of plasma. We have added this clustering analysis to the manuscript and modified the text accordingly. Future studies will include significantly more patients, minimizing the variation in sample collection and timing that resulted from increasing our clinical co-hort.

Figure 4a and 4b. Unsupervised clustering of the 100 most variant genes identified in the EVs of patients and healthy controls using the ^{EV}HB-Chip. For plasma (**a**), data shows 10 GBM patients clustering together with one patient clustering outside the group, Patient 12, which was later determined to be an anaplastic oligodendroglioma

tumor (AO). For serum (b) GBM patients clustered together and separated from healthy controls.

4. If the expression profile of exosome vesicle contained the features for four subtypes, authors can classify the patients into subtypes based on their exosome vesicle expression patterns.

R. The cancer genome atlas network has previously classified glioblastomas based on levels of sets of gene expression: Proneural, neural, classical, and mesenchymal.¹² These subtypes indicate the remarkable heterogeneity of GBM.¹³ However, a recent report of single cell analysis of solid GBM tumors suggest that individual tumors have heterogeneous mixtures of cells that correspond to different glioblastoma subtypes and hybrid cellular states.¹⁴ Our results showing EVs from different patients with different characteristic subtype genes support those findings, and indicating that the heterogeneity of the disease can be potentially recapitulated with RNAseq of EVs. Therefore, we are not trying to classify the patients by subtype. Instead we are trying to emphasize that the heterogeneity implies a more complex diagnostic and treatment. Lastly, as much as we would like to see if our analysis of the EVs aligns with the primary tumor sub-type, we were informed that subtyping analysis is still not standard for patients and was not done for the patients that we evaluated.

5. Can you explain why Cetuximab only is better than cocktail antibody to capture the exosome vesicle?

R. Please see our response to comment 1 of Reviewer 2 as a similar concern was raised.

We thank all of the reviewers for their input and time; we look forward to your response to our revision.

References

1. Zhao, C. & Cheng, X. Microfluidic separation of viruses from blood cells based on intrinsic transport processes. *Biomicrofluidics* **5**, 032004-032010 (2011).
2. Sin, A., Murthy, S.K., Revzin, A., Tompkins, R.G. & Toner, M. Enrichment using antibody-coated microfluidic chambers in shear flow: Model mixtures of human lymphocytes. *Biotechnol. Bioeng.* **91**, 816-826 (2005).
3. Cheng, X., *et al.* A microfluidic device for practical label-free CD4+ T cell counting of HIV-infected subjects. *Lab Chip* **7**, 170-178 (2007).
4. Usami, S., Chen, H.-H., Zhao, Y., Chien, S. & Skalak, R. Design and construction of a linear shear stress flow chamber. *Ann. Biomed. Eng.* **21**, 77-83 (1993).
5. Shao, H., *et al.* Chip-based analysis of exosomal mRNA mediating drug resistance in glioblastoma. **6**, 6999 (2015).
6. Schneider, C.A., Rasband, W.S. & Eliceiri, K.W. NIH Image to ImageJ: 25 years of image analysis. *Nat Meth* **9**, 671-675 (2012).
7. Chen, C., *et al.* Microfluidic isolation and transcriptome analysis of serum microvesicles. *Lab Chip* **10**, 505-511 (2010).
8. A, M., A, S. & G, U. The role of EpCAM in physiology and pathology of the epithelium. *Histology and Histopathology* **31**, 349-355 (2016).
9. Tauro, B.J., *et al.* Comparison of ultracentrifugation, density gradient separation, and immunoaffinity capture methods for isolating human colon cancer cell line LIM1863-derived exosomes. *Methods* **56**, 293-304 (2012).
10. He, M., Crow, J., Roth, M., Zeng, Y. & Godwin, A.K. Integrated immunoisolation and protein analysis of circulating exosomes using microfluidic technology. *Lab Chip* **14**, 3773-3780 (2014).
11. Zhao, Z., Yang, Y., Zeng, Y. & He, M. A microfluidic ExoSearch chip for multiplexed exosome detection towards blood-based ovarian cancer diagnosis. *Lab Chip* **16**, 489-496 (2016).
12. Verhaak, R.G.W., *et al.* An integrated genomic analysis identifies clinically relevant subtypes of glioblastoma characterized by abnormalities in PDGFRA, IDH1, EGFR and NF1. *Cancer Cell* **17**, 98 (2010).
13. Sottoriva, A., *et al.* Intratumor heterogeneity in human glioblastoma reflects cancer evolutionary dynamics. *Proc. Natl. Acad. Sci.* **110**, 4009-4014 (2013).
14. Patel, A.P., *et al.* Single-cell RNA-seq highlights intratumoral heterogeneity in primary glioblastoma. *Science (New York, N.Y.)* **344**, 1396-1401 (2014).

REVIEWERS' COMMENTS:

Reviewer #1 (Remarks to the Author):

The authors have addressed the last set of reviews and I am satisfied that the paper has improved even further and should be published.

Reviewer #2 (Remarks to the Author):

The authors have addressed all of my comments and the manuscript is now suitable for publication.

Reviewer #3 (Remarks to the Author):

All the questions were clearly answered with additional data and description by authors.